# Execution and Design of an Anti HPIV-1 Vaccine with Multiple Epitopes Triggering Innate and Adaptive Immune Responses: An Immunoinformatic Approach

**DOI:** 10.3390/vaccines10060869

**Published:** 2022-05-29

**Authors:** Muhammad Naveed, Allah Rakha Yaseen, Hira Khalid, Urooj Ali, Ali A. Rabaan, Mohamed Garout, Muhammad A. Halwani, Abbas Al Mutair, Saad Alhumaid, Zainab Al Alawi, Yousef N. Alhashem, Naveed Ahmed, Chan Yean Yean

**Affiliations:** 1Department of Biotechnology, Faculty of Life Sciences, University of Central Punjab, Lahore 54000, Pakistan; 123allah.rakha@gmail.com (A.R.Y.); msurooj9@gmail.com (U.A.); 2Department of Medical Education, King Edward Medical University, Lahore 54000, Pakistan; hirakhalid6001@gmail.com; 3Molecular Diagnostic Laboratory, Johns Hopkins Aramco Healthcare, Dhahran 31311, Saudi Arabia; 4College of Medicine, Alfaisal University, Riyadh 11533, Saudi Arabia; 5Department of Public Health and Nutrition, The University of Haripur, Haripur 22610, Pakistan; 6Department of Community Medicine and Health Care for Pilgrims, Faculty of Medicine, Umm Al-Qura University, Makkah 21955, Saudi Arabia; magarout@uqu.edu.sa; 7Department of Medical Microbiology, Faculty of Medicine, Al Baha University, Al Baha 4781, Saudi Arabia; mhalwani@bu.edu.sa; 8Research Center, Almoosa Specialist Hospital, Al-Ahsa 36342, Saudi Arabia; abbas.almutair@almoosahospital.com.sa; 9College of Nursing, Princess Norah Bint Abdulrahman University, Riyadh 11564, Saudi Arabia; 10School of Nursing, Wollongong University, Wollongong, NSW 2522, Australia; 11Administration of Pharmaceutical Care, Al-Ahsa Health Cluster, Ministry of Health, Al-Ahsa 31982, Saudi Arabia; saalhumaid@moh.gov.sa; 12Division of Allergy and Immunology, College of Medicine, King Faisal University, Al-Ahsa 31982, Saudi Arabia; zalalwi@kfu.edu.sa; 13Department of Clinical Laboratory Sciences, Mohammed AlMana College of Health Sciences, Dammam 34222, Saudi Arabia; yousefa@machs.edu.sa; 14Department of Medical Microbiology and Parasitology, School of Medical Sciences, Universiti Sains Malaysia, Kubang Kerian, Kota Bharu 16150, Malaysia; namalik288@gmail.com

**Keywords:** laryngotracheobronchitis, HPIV Type-I, hemagglutinin-neuraminidase (HN) protein, computational vaccine, molecular docking, immune simulations

## Abstract

Human Parainfluenza Virus (HPIV) Type-1, which is an anti-sense ribonucleic acid (RNA) virus belonging to the paramyxoviridae family, induces upper and lower respiratory tract infections. The infections caused by the HPIV Type-1 virus are usually confined to northwestern regions of America. HPIV-1 causes infections through the virulence of the hemagglutinin-neuraminidase (HN) protein, which plays a key role in the attachment of the viral particle with the host’s receptor cells. To the best of our knowledge, there is no effective antiviral drugs or vaccines being developed to combat the infection caused by HPIV-1. In the current study, a multiple epitope-based vaccine was designed against HPIV-1 by taking the viral HN protein as a probable vaccine candidate. The multiple epitopes were selected in accordance with their allergenicity, antigenicity and toxicity scoring. The determined epitopes of the HN protein were connected simultaneously using specific conjugates along with an adjuvant to construct the subunit vaccine, with an antigenicity score of 0.6406. The constructed vaccine model was docked with various Toll-like Receptors (TLRs) and was computationally cloned in a pET28a (+) vector to analyze the expression of vaccine sequence in the biological system. Immune stimulations carried out by the C-ImmSim Server showed an excellent result of the body’s defense system against the constructed vaccine model. The AllerTop tool predicted that the construct was non-allergen with and without the adjuvant sequence, and the VaxiJen 2.0 with 0.4 threshold predicted that the construct was antigenic, while the Toxinpred predicted that the construct was non-toxic. Protparam results showed that the selected protein was stable with 36.48 instability index (II) scores. The Grand average of Hydropathicity or GRAVY score indicated that the constructed protein was hydrophilic in nature. Aliphatic index values (93.53) confirmed that the construct was thermostable. This integrated computational approach shows that the constructed vaccine model has a potential to combat laryngotracheobronchitis infections caused by HPIV-I.

## 1. Introduction

HPIVs are non-segmented and enclosed viruses, having anti-sense RNA that infect the respiratory tract, especially its ciliated epithelial cells [1], of any person of any age [2,3,4,5]. As reported, the HPIV genome contains more than 15,000 nucleotides which encode six different structural proteins [6]. HPIVs are the most common pathogens that cause seasonal respiratory tract infections; the first three types mostly affect infants, young children, and immuno-compromised persons causing lower respiratory tract infections (LRTIs) with conditions such as bronchiolitis, mild cold to croup, and viral pneumonia [4,7,8].

In the United States of America (USA), HPIV-1 is proved to be the most common reason for laryngotracheobronchitis or simply known as croup, which is the reason for the hospitalization of more than 30,000 victims every year in the USA [9,10,11]. In Pakistan, however, incidents of HPIV have not been evaluated since 1985, whereby HPIV-1 and HPIV-3 were recorded as the most pathogenic [12]. Although HPIV outbreaks are widely studied in Western nations, they remain underexplored and are not included in the usual laboratory confirmation in many Asian countries. Adding to it, there is no effective vaccine available to prevent the infection. However, by using a multi-disciplinary approach, a study reported on suramin, a drug that inhibits the hemagglutinin-neuraminidase of HPIV by non-competitive interaction and slows down replication of the virus in mammalian host cells [9].

To overcome or more specifically reduce any kind of allergic reactions, researchers are adapting to peptide-based vaccine designed by the use of recent advancements in computational biology [13,14]. Vaccines based on T and B cell-specific epitopes are designed to activate the immune system of the person to protect its body against possible viral attacks in the future; for designing such vaccines, immunoinformatic contributes crucially in data analysis and management using computer-aided research strategies with high-confidence predictions [15,16,17].

The establishment of epitope-based vaccines involves the attachment of predicted epitopes and adjuvant through linkers. In epitope-based vaccines, the antigenic effect is limited as compared to live attenuated vaccines. In case of a purified epitope vaccine, the adjuvant is an essential component that can enhance the magnitude of antigenicity of antigens (epitopes) in adaptive immune responses. This magnification of antigens by an adjuvant can enhance antibody production and affects the longevity of the vaccine through the co-administration of epitopes. A limited number of reported adjuvants are available that are incorporated into human vaccines [18]. Here, a widely acclaimed immunoinformatic approach was pertained to constructing a vaccine against HPIV-1 targeting the host’s immune cells using available molecular data about the virus.

To the best of our knowledge, there is no vaccine of HPIV-1 using HN protein that has been reported yet. Epitope-based vaccines hold promise over other types because of their ability to induce a substantial immune response in a lower concentration. The current study was designed to identify the highly antigenic peptides having a tendency to induce the immune cells (B cells; Th and TC cells) and interferons. The construct was screened for antigenicity, allergenicity, and physicochemical properties along with protein–protein docking with TLR receptors. Immune simulations were performed to predict vaccine efficiency.

## 2. Materials and Methods

### 2.1. Viral Proteins’ Sequence Retrieval

The FASTA format sequences of fusion glycoprotein (F), membrane (M) protein, hemagglutinin-neuraminidase (HN) protein, nucleocapsid (N) protein, and nucleoprotein of several HPIV-1 strains were fetched from the UniPort database (https://www.uniprot.org/ (accessed on 25 July 2021)) having UniProt IDs P12605, Q01427, P16071, J7FXM0, and P26590, respectively. VaxiJen score of five selected proteins along with their UniProt IDs has been mentioned in the Appendix A. While the flow chart of methodology has been shown in Appendix A.

### 2.2. Antigenic Protein Identification

The retrieved proteins were screened for vaccine candidacy on the VaxiJen v2.0 server to evaluate their antigenicity values [19]; the server is reachable at http://www.ddg-pharmfac.net/vaxijen/ (accessed on 25 July 2021).

### 2.3. Physicochemical Analysis

The physicochemical characteristics of the selected protein were computed via ProtParam, which is available at the ExPasy server [20]. The parameters calculated by ProtParam (https://web.expasy.org/protparam/ (accessed on 25 July 2021) include but are not limited to atomic composition, amino acid composition, instability and aliphatic index, extinction coefficient, Grand average of Hydropathicity (GRAVY), and theoretical PI.

### 2.4. Two-Dimensional (2D) and 3D Structural Analysis

Secondary structure analyses of the protein were rendered by PSIPRED [21], while I-TASSER [22] provided the 3D model of the protein according to the highest C-Score value.

### 2.5. B-Cell-Specific Epitopes Identification via IEDB Tools

The B-cell Antigen Sequence Properties tool of IEDB [23] (http://tools.iedb.org/bcell/ (accessed on 26 July 2021) calculated the linear epitopes confined to B cells. The Bepipred Linear Epitope Prediction 2.0 method of prediction was selected with a threshold of 0.500 and 4 center positions; this method predicts B-cell epitopes from a protein sequence, using a Random Forest algorithm trained on epitopes and non-epitope amino acids. It determines the results from crystal structures, which follows by the sequential prediction later.

### 2.6. Predicting T-Cells Restricted Epitopes

Tools of the IEDB T-cell database were utilized to analyze the epitopes restricted by both types, type I (http://tools.iedb.org/mhci/ (accessed on 26 July 2021) and II (http://tools.iedb.org/mhcii/ (accessed on 26 July 2021) of the major histocompatibility complex [24].

### 2.7. Selected Epitopes’ Population Coverage

To check which epitopes cover the maximum world population, we used the population coverage analysis integrated in the IEDB server [25], which is provided at http://tools.iedb.org/population/ (accessed on 27 July 2021). In this tool, interaction of the selective alleles of the major histocompatibility complex with the selected epitopes was examined, screening the ones that cover the maximum population (most number of alleles).

### 2.8. Profiling of Predicted Epitopes

The finalization of the suitable epitopes was completed by further filtration of predicted epitopes. To this end, Toxinpred analysis (https://webs.iiitd.edu.in/raghava/toxinpred/algo.php (accessed on 27 July 2021) [26]), VaxiJen v2.0 [27], and AllerTOP [28], available at https://www.ddg-pharmfac.net/AllerTOP/method.html (accessed on 27 July 2021), were employed for toxicity, antigenicity, and allergenicity profiling sequentially.

### 2.9. Vaccine Design and Profiling

A vaccine construct consists of four main components namely, an adjuvant, linkers, B and T cell epitopes, and 6X histidine molecules. All epitopes with the target were linked using specific linkers along with adjuvant Human Beta Defensin 3 with UniProt ID: Q5U7J2 [29]. Antigenic and allergenic profiling of the construct was executed by the implementation of Allertop and Vaxijen (version 2.0 for both), while ExPasy ProtParam examined the physicochemical attributes of the assembled vaccine.

### 2.10. Disulfide Engineering of Vaccine

The Disulfide tool on Design 2.0 (DbD2) was employed to perform the Disulfide concoction of the constructed vaccine [30]; this tool helped in designing novel disulfide bonds in the constructed vaccine.

### 2.11. Three-Dimensional (3D) Structural Analysis

RaptorX was employed to investigate the tertiary structure of the fabricated vaccine [31] (http://raptorx.uchicago.edu/ (accessed on 28 July 2021). ProSA-web, introduced by [32], provided at https://prosa.services.came.sbg.ac.at/prosa.php (accessed on 28 July 2021), and Ramachandran plot assessment was executed via PROCHECK v6.0 [33] (https://saves.mbi.ucla.edu/ (accessed on 28 July 2021), for the validation and stability confirmation of the constructed vaccine model. GalaxyRefine [34] available at (http://galaxy.seoklab.org/cgi-bin/submit.cgi?type=REFINE (accessed on 28 July 2021) refined the 3D vaccine model.

### 2.12. Molecular Docking and Dynamics

Molecular docking was performed by online docking servers, i.e., ClusPro (https://cluspro.bu.edu/login.php?redir=/home.php (accessed on 29 July 2021) [35] and PatchDock [36] (https://bioinfo3d.cs.tau.ac.il/PatchDock/php.php (accessed on 29 July 2021). The refined vaccine model and different Toll-like receptors (TLR2 with PDB ID: 6nig, TLR3 with PDB ID: 2a0z, TLR4 with 4g8a, TLR5 with PDB ID: 3j0a, and TLR8 with 4r0A) were docked using both online servers. TLRs are the main human body receptors recognizing and binding to various epitopes to generate an immune response against them. The vaccine was docked against these receptors to evaluate whether it fits in the binding pocket and interacts enough to elicit a response. Molecular dynamic simulations of the docked complexes were performed using the iMODS server [37].

### 2.13. Codon Improvement and Cloning

The Java Codon Adaptation Tool was applied to perfect the codons of refined vaccine construct for enhanced expression [38] (https://www.jcat.de/ (accessed on 31 July 2021), and in silico cloning was performed after optimization through the SnapGene software introduced by Insightful Science (snapgene.com (accessed on 31 July 2021)).

### 2.14. Immune Simulation

The C-IMMSIM server produced immune simulations (https://kraken.iac.rm.cnr.it/C-IMMSIM/ (accessed on 31 July 2021) [39], as it estimates the immunological response to the proposed immunization construct.

## 3. Results

### 3.1. Antigenicity, Structural and Physicochemical Analysis of Viral HN Protein

The antigenicity scores-based analysis of all the candidate proteins showed that the HN protein was a viable candidate for vaccine construction, and the physicochemical parameters of selected protein are given in Table 1. The HN protein contains 575 amino acids weighing 63,981.34 Da. Protparam results revealed that the selected protein is stable with instability index (II) scores of 36.48. The tertiary structure of the HN protein is shown in Figure 1A. It contains 10.78% α helixes, 30.26% β sheets, and 58.96% loops (Figure 1C). The residues from 42 to 60 were found to be inside the transmembrane region, and residues from 1 to 41 were found to be in the cytoplasmic region, and the core-region of the HN protein is extracellular (Figure 1B,D). Ramachandran plot predicted by the PROCHECK using the refined model of vaccine has been shown in Appendix A. While the Graphical representation of Z-score calculation of Peptide compared with the native protein energies has been shown in Appendix A.

### 3.2. B-Cell-Restricted Epitope Predictions

Residues of the HN protein having scores above the threshold of 0.500 of the tools (IEDB’s Bepipred Linear Epitope Prediction 2.0 method) were predicted to be a member of an epitope. The tool predicted a total number of 19 B-cell epitopes, but for further analysis, only the epitopes with the length of 9–50 amino acids were picked. B-cell-restricted epitope predictions of HN protein objectively allowed the finding of potential antigens that would interact efficiently with B-lymphocytes. B-cell-restricted epitopes of HN protein will generate rich immune responses against viral infection upon recognition by the surface receptors of B-lymphocytes. The stimulation by these epitopes will trigger the differentiation of B-cells to plasma cells, generating the antibodies able to recognize the epitopes of the viral HN protein. The Appendix A shows the B-Cell epitopes predicted by the IEDB server along with the length of each epitope.

### 3.3. Predicting T-Cells Restricted Epitopes

More than 30,000 possible MHC-I binding epitopes were predicted; out of these, the top 50 epitopes, based on the IC50 value, were filtered and analyzed for the vaccine construct. Similarly, epitopes from the viral core peptides binding to MHC-II molecules were selected and analyzed. The prediction of epitopes restricted to MHC-I and MHC-II using the viral HN protein is crucial, as the MHCs (more specifically HLAs in humans) are the coordinators between innate and adaptive immune defense, which help in response transition. Both classes, MHC-I and MHC-II, are restricted to CD8+ cytotoxic T-lymphocytes (CTLs) and CD4+ helper T-lymphocytes, respectively, and both play a vital role in viral infections. The antigen-presenting cells (APCs) from the innate immune response deliver the viral epitopes to MHC molecules, which activate the adaptive immune response, i.e., activation of B-cells and production of antibodies. T-cell-restricted epitopes of HN protein will just stimulate the MHC molecules available at T-cells and create immunity against HPIV-1 viral infection. Reference set alleles for MHC-1 and MHC-II molecules has been shown in Appendix A.

### 3.4. Filtration of Predicted Epitopes

Out of 19, only three epitopes of the B-cell were picked to become part of the final vaccine construct; similarly, 16 epitopes restricted by MHC-I and 15 epitopes for MHC-II passed the filtration tests and were selected for the construct. All epitopes that passed the filters are given in Table 2 and Table 3.

### 3.5. Population-Coverage Analysis

A cumulative coverage of the selected epitopes illustrated 98.55% and 72.18% against MHC-I and II molecules sequentially, advocating the efficiency of the construct. The individual epitope coverage for each MHC class is depicted in Figure 2. Coverage of some alleles was not available for computation, and therefore, they were omitted from the analysis.

### 3.6. Assembly of a Multi-Epitope Vaccine

The assembly started by joining all the B-cell epitopes to the adjuvant sequence using EAAAK linkers, which were then connected to the MHC-I epitopes, using the GPGPG linkers. Furthermore, MHC-II was linked to the chain via AYY linkers. The 6xHis tag was utilized at the carbon and the amino-terminus of the sequence for protein identification and purification during the cloning process. The construct is provided in Figure 3.

### 3.7. Quality Check Analysis and Profiling of Designed Vaccine

AllerTop predicted the construct is non-allergen with and without the adjuvant sequence, and the VaxiJen 2.0 predicted that the construct is antigenic with 0.6406 and 0.6510 scores, respectively, with and without the adjuvant. According to Toxinpred predictions, the construct was evaluated to be non-toxic in nature.

### 3.8. Physicochemical Properties

The construct composed of 641 amino acids having 69661.70 Da of molecular weight, and the molecular formula exhibited the exact number of atoms present in the vaccine peptide, while the other physicochemical properties are shown in Table 1. Protparam results showed that the constructed protein was highly stable with a 33.32 instability index score, indicating that it can initiate a viable immune response with adequate dosage. A theoretical pI of 9.04 suggested a good structural fold of the vaccine helping with the vaccine purification methods. According to the Grand average of Hydropathicity scores, the constructed protein is hydrophilic, allowing it to interact with water residues in the host cells promoting its immunogenicity. The high aliphatic index (93.53) proves the thermostability of the vaccine, indicating that it can tolerate sudden temperature changes effectively. The short half-life of the construct (3.5 h), evidenced in vitro, depicts that the antigen remains in the host for a short span that is enough to induce the host’s immune response but not long enough to be pathogenic.

### 3.9. Prediction of Secondary and Tertiary Structure

Raptor X demonstrated that the protein has 34% helices, 4% beta sheets, and 61% coils. More than half (59%) of the protein residues were categorized as exposed (E), 19% were found to be medium exposed (M), and 20% of the residues were considered buried (B). There was a total of 50 residues, constituting 7% of total protein content, that were found to be disordered. The three-dimensional vaccine construct with the Highest CNS-solve program score was further processed for refinement and validation analysis.

### 3.10. Tertiary Structure Refinement and Validation

The GalaxyRefine server considered different parameters in the refinement process. It was noted from the Ramachandran plot that 90.5% residue of the refined protein was in the highly acceptable region, 8.3% residues were found in the moderately allowed region, 0.8% were present in the scarcely allowed sector of the plot, and 0.4% residues were in the outlier or unfavorable region. The predicted z-score was −5.06, validating the refined 3D structure. Scores of different parameters used by GalaxyRefine for the refinement of 3-D model has been shown in Appendix A.

### 3.11. Vaccine Stability by Disulfide Engineering

The disulfide engineering was completed using a DbD2 server between the 12 selected residue pairs, selected on the basis of energy level between them. The mutated structure with possible predicted disulfide bonds in yellow color is given in Appendix A, whereas Appendix A showed the original structure (wireframe structure). The list of 12 selected residue pairs for disulfide engineering has been shown in Appendix A.

### 3.12. Molecular Docking of Designed Vaccine with Ligand Binding TLRs

Figure 4A–E represent the docked structures of the vaccine construct and the chosen TLRs. Out of the 30 models depicted for all the complexes, the TLR2 + vaccine model with −1403.6 kCal/mole binding energy (Figure 4A), TLR3 + vaccine model with −1237.9 kCal/mole energy (Figure 4B), TLR4 + vaccine model with −1258.0 kCal/mole energy (Figure 4C), TLR5 + vaccine model with −1832.9 kCal/mole energy (Figure 4D), and TLR8 + vaccine model with −1475.6 kCal/mole energy (Figure 4E) were selected. The selection of models was made based on the lowest binding energy (highest negative energy), as it is indicative of excellent binding affinity, since greater negative energy stabilizes the reaction toward the product (in our case, the vaccine and the TLR complex). These calculations suggested that the vaccine construct has the strongest interaction with TLR5, allowing this receptor to take part in immune response initiation most influentially; however, the other receptors also generate a suitable immune response. The maximum ClusPro energies, cluster members and patchdock of all the TLRs used for Molecular docking has been shown in Appendix A.

### 3.13. Molecular Dynamic Simulations

MD simulations results were almost immediately available for the provided complex (TLR2; Vaccine). The models revealed little contortion at each residue’s capacity level. The NMA-calculated B-factor scores were the same as the RMS values. An elasticated network model, shown in Figure 5, was developed to detect the pairings of atoms linked with springs. Every dot in the figure represented a spring in between respective atomic pairs and was shaded based on its rigidity. The springs became firmer as the grays became darker. The goal of ENM (depicted in Figure 5) is to linearize the complex motions of a macromolecule (in our case, the vaccine: TLR complex). It helps understanding the atomic motions of the complex. As discussed above, the darker-gray dots represent stiff atoms corresponding to limited motion and reflecting the fact that the conformation of the complex is stable. The 3-D model, B-factor/Mobility, Eigenvalues, Variance, Covariance map, and Elastic network of TLR-2, TLR-3, TLR-4, TLR-5 and TLR-8 Complex has been shown in Appendix A respectively.

### 3.14. Adaptation and Enhancement of Codon along with In-Silico Cloning

The refined codon series was 1923 nucleotides in length, with a CAI of 0.97 and a GC content of 53.25% (Figure 6). The improved nucleotide sequence was integrated into the pET28a (+) plasmid after being replicated in silico by employing SnapGene software to generate a recombinant plasmid (Figure 7).

### 3.15. Immune Simulations

The vaccine was simulated with three doses with a gap of 4 weeks and 1050 Simulation Steps were adjusted; these settings gave the elapsed time of 349 days. The result showed an increase in B-cells along with a rise in IgM and IgG antibodies, as shown in Figure 8A,B. The primary and secondary response also represented the rise in T-cell (T_H_, T_R_, T_C_) population within 50 days and an increase in the NK cell population (Figure 9A,B). Figure 10 shows peaks representing the production of different immune molecules such as IFN-g, IL4, IL-12, IFN-a, IL-2, etc. The production pattern of NK cells has been shown in Appendix A.

## 4. Discussion

The innovation in computational biology helps the researchers search for new drugs and vaccines [39]. The primary objective of current study was to construct libraries of all the possible epitopes of B-cell and MHC class I and II of T-cells with the most interacting human leukocyte using the IEDB server (with reference HLA alleles) and to filter those epitopes by the fixed criteria to select the best possible residue to become part of the vaccine construct. Then, the selected epitopes were joined together via linkers (EAAAK, GPGPG, AYY). In the next step, the construct was checked for allergenicity using AllerTop and for antigenicity using VaxiJen.

Designing novel vaccines is much needed to immunize human beings against such seasonal epidemic diseases [40,41]. Improvement in bioinformatics tools, rapid advancement in biotechnology, and a vast understanding of immunology helped researchers to develop new methods to design vaccines [42]. Bioinformatics tools, databases, and programs can predict epitope peptides, which can act as ligands to initiate immune responses [43]. The identification of potential epitopes (peptides of pathogen) for B-cells and T-cells using an immunoinformatic approach can lead to designing a peptide-based vaccine to induce immunity against a specific pathogen with higher efficacy and fewer side effects. Bioinformatics tools and databases can predict epitope-based peptides, which can act as ligands to initiate the immune response [44]. Epitope-based vaccines against Fowlpox Virus [45], Hepatitis B Virus [46], Nipah Henipavirus [47], and novel SARS-CoV-2 had been reported using the Immunoinformatics approach [42].

Several reports and a few clinical trials suggested the spike protein of any virus [17,48], such as the HN protein of HPIV-1, can be used for epitope vaccine designing [49]. Some studies targeted the sequences of more than one viral protein together to predict the B-cell and T-cell epitopes [29,42]; on the other hand, some researchers target a single protein to design the in silico vaccines [50,51]. The reported therapeutics targeting the glycoproteins of HPIV-1 envelope is the use of suramin, a non-competitive inhibitor of HN protein [52], and the utilization of monoclonal antibodies to neutralize the hemagglutinin-neuraminidase [53]; both strategies involve the inhibition of fusion protein activity by deactivating and neutralizing the HN protein. This study focuses on the same protein; the secondary structure and physicochemical analysis of the targeted protein show it to be highly antigenic, making it a great candidate for multi-epitope vaccine designing.

MHC-I and MHC-II are involved in an adaptive immune response where these peptides present the antigen to CD4+ (cytotoxic T-cells) and CD8+ (helper T-cells), respectively [54]. Meanwhile, the B-cell epitopes are recognized by the B-lymphocytes (via B-cell receptors) [55]. So, the current study utilizes the combination of MHC-I and MHC-II T-cell restricted epitopes and B-cell cell restricted epitopes of the HN protein for the development of strong immunity.

The selected epitopes in this study were combined together via linkers (i.e., EAAAK, GPGPG, AYY) as discussed in a previous study by Naveed, Tehreem, Arshad, Bukhari, Shabbir, Essa, Ali, Zaib, Khan, Al-Harrasi and Khan [48]; the construct also contains an adjuvant with the purpose to increase the immunoreactive property of the vaccine as put forward by Tahir Ul Qamar et al. [56]. Several studies have reported different adjuvants in epitope vaccine designs including the 50S ribosomal protein L7/L12 with UniProt ID: P9WHE3 [48], HABA Protein with accession number: AGV15514.1, and Human Beta Defensin 1, 2, 3 [57]. The vaccine construct in the current study includes Human Beta Defensin 3 with UniProt ID: Q5U7J2, because other adjuvants such as 50S ribosomal protein L7/L12 and HABA Protein reduced the antigenicity score (0.5984 and 0.5668, respectively) and increased the construct size from the ideal length.

The vaccine 3D structure predicted, refined and validated so we can perform docking with TLRs, because TLRs play an important role in the initiation of innate immune response [58]. TLRs are essential receptor proteins that recognize pathogens through Pathogen-Associated Molecular Patterns (PAMPs) and activate immune response by interacting with pathogens’ nucleic acid and envelope proteins [58]. Hence, this study focused on the interaction between the vaccine construct and five different TLRs (TLR 2, TLR 3, TLR 4, TLR5, and TLR 8). According to Gibney et al. [59], various interactions, i.e., electrostatic, Van der Waals, hydrophobic and hydrophilic, describe the stable configuration of complexes in docked clusters, which were mirrored here. The B-factor/mobility, variance, covariance analysis, and elastic network of the complexes supported the fact that the vaccine construct tends to initiate the humoral immune response as reported by Ali et al. [60].

**Study limitations:** The current study was performed in silico, and no wet lab experiments were carried out to empower the reported outcomes. Although excellent results were observed in the B-cell and T-cell epitope analysis, problems such as the non-availability of most HLAs in the IEDB, the 9–14 AA-specific immune response that may lead to variable results when tested in vivo and in vitro due to length restriction, the poorly defined motifs in the epitopes, and the TLR–vaccine complex could not be addressed in the study. It is therefore recommended for future researchers to find alternative for these limitations, especially providing wet-lab evidence wherever possible.

## 5. Conclusions

HPIV-1 is not catered by any of the existing preventive measures, leading to limits regarding the consistent increase in its incidences. The current study construct provides a clinical breakthrough in response against HPIV infections with hope to limit its spread and pathogenesis by designing an epitope-based vaccine using the HN protein as an immunogenic target by an immunoinformatic approach. The protein–protein interaction showed that the construct has the potential to be recognized by the immune cell receptors. The cloning of a construct in the PET28a (+) plasmid using snapgene showed that the expression level of the vaccine peptide was excellent. Immune simulations illustrated that this construct can generate a good immune response by the significant production of T-cells, B-cells, and other immune molecules. This integrated computational approach saves time, is cost effective, and guides experiments with less chance of errors and with fewer trials. However, the efficacy of any drug or vaccine can be ensured via experimental validations (both in vitro and in vivo). At clinical trials levels, the other peptide vaccines had shown good results with much better immune response, so the current vaccine construct could be considered a good candidate against Human Parainfluenza Virus Type-1.

## Figures and Tables

**Figure 1 vaccines-10-00869-f001:**
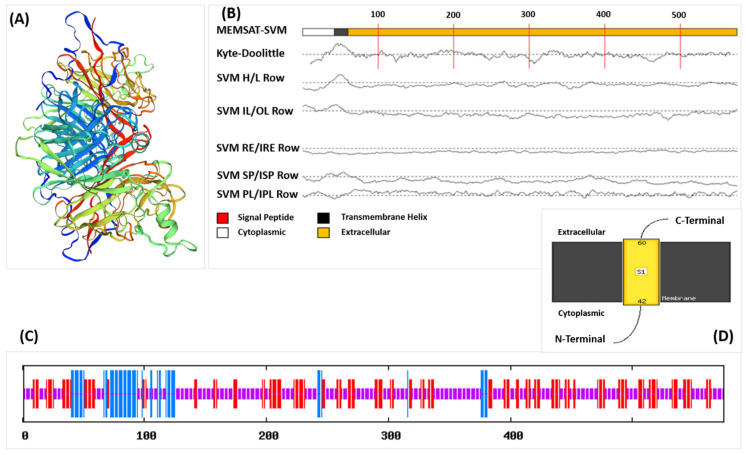
(**A**) Three-dimensional (3D) structure of the HN protein obtained from PDB, (**B**) MEMSAT predicted by PsiPred showing the Topology of the HN protein, (**C**) Secondary structure of HN protein predicted by SOPMA showing different components in different colors, (**D**) MEMSAT showing extracellular part of HN protein.

**Figure 2 vaccines-10-00869-f002:**
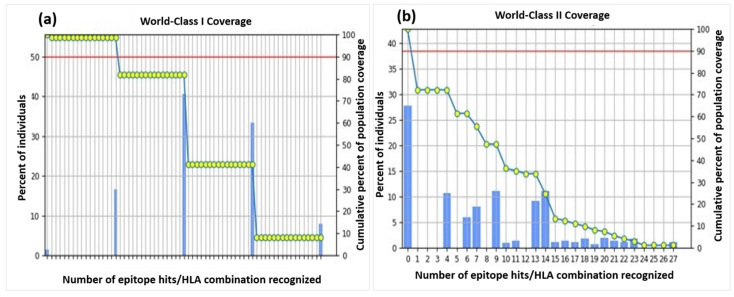
Population coverage of all the selected epitopes for both MHC-II classes. (**a**) World-Class I Coverage. (**b**) World-Class II Coverage.

**Figure 3 vaccines-10-00869-f003:**
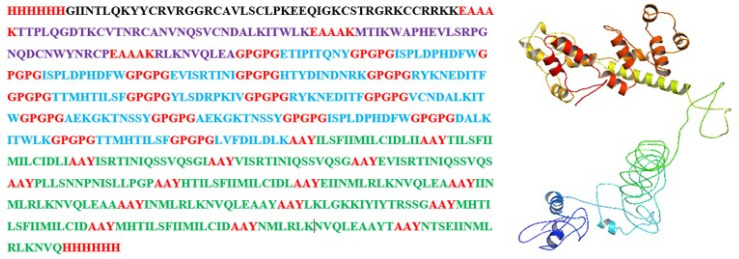
Prepared vaccine construct with the His-tag and linkers in red, adjuvant in black, B-cell epitopes in purple, MHC-I epitopes in blue, and MHC-II epitopes in green and its 3D model.

**Figure 4 vaccines-10-00869-f004:**
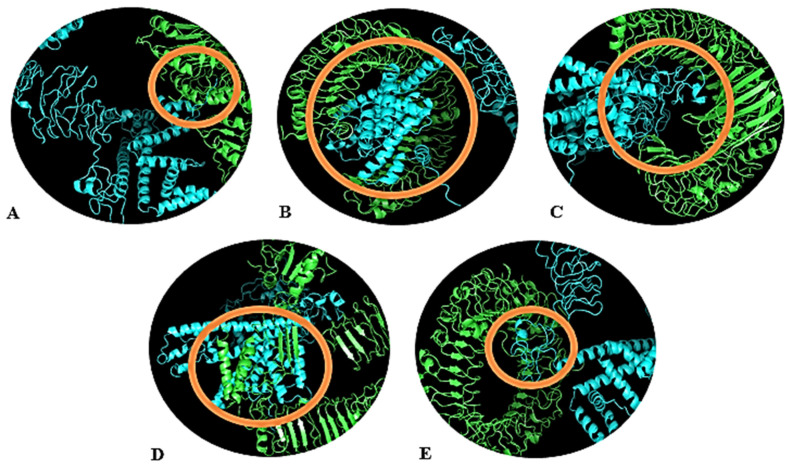
Docked complexes of the vaccine and the TLRs. (**A**) TLR-2, (**B**) TLR-3, (**C**) TLR-4, (**D**) TLR-5, (**E**) TLR-8. The TLRs are shown in green color, whereas the vaccine construct is in blue color. Interacting residues are highlighted with an orange circle.

**Figure 5 vaccines-10-00869-f005:**
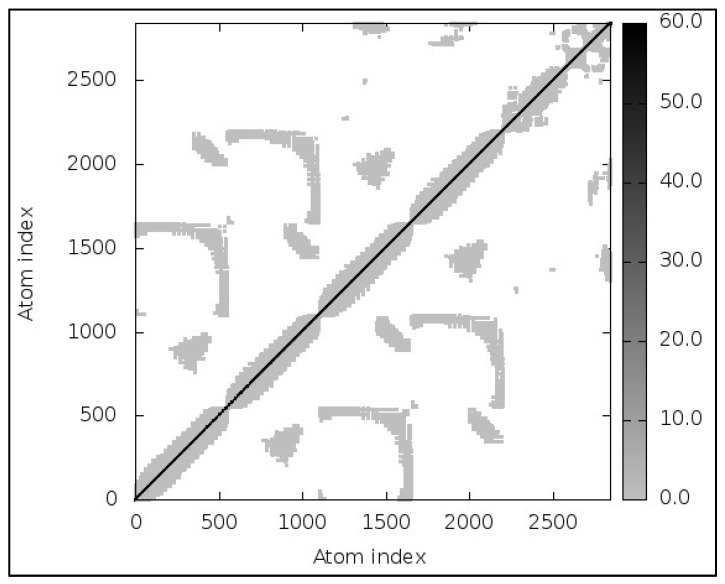
Elastic network of TLR-2: vaccine complex.

**Figure 6 vaccines-10-00869-f006:**
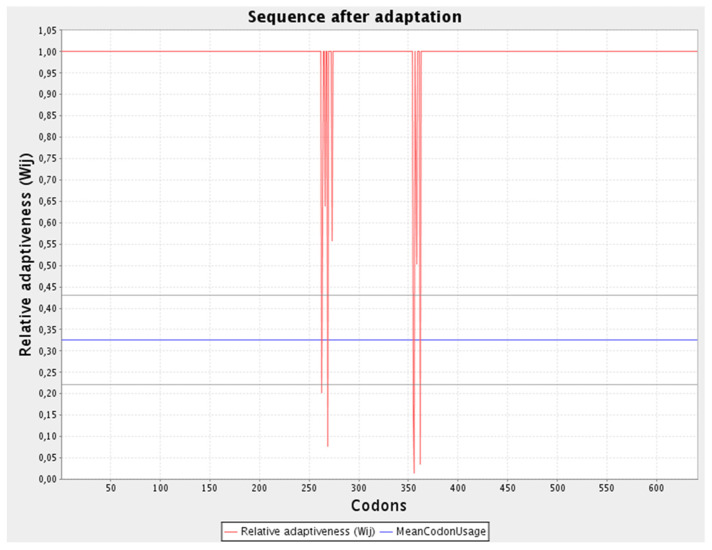
Sequence adaptation predicted via JCAT.

**Figure 7 vaccines-10-00869-f007:**
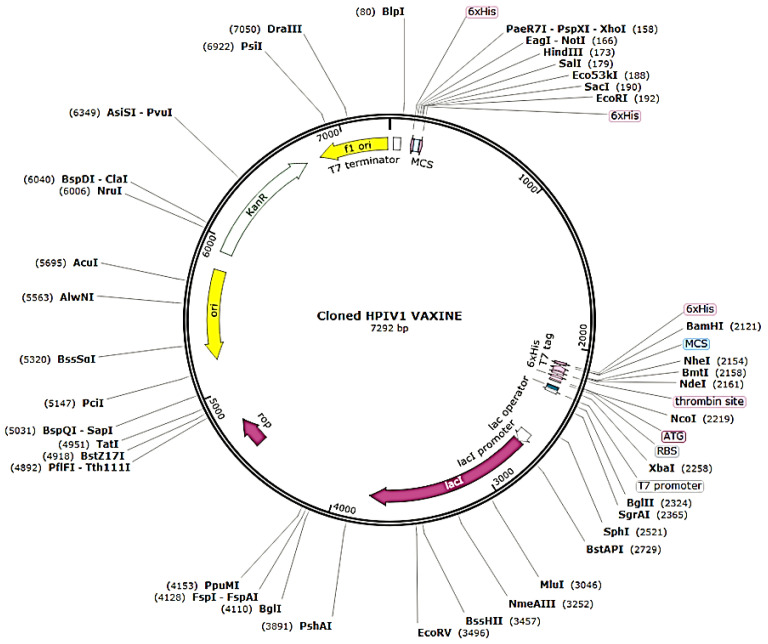
In silico replication of the pET28a (+) plasmid using SnapGene software to construct a recombinant plasmid.

**Figure 8 vaccines-10-00869-f008:**
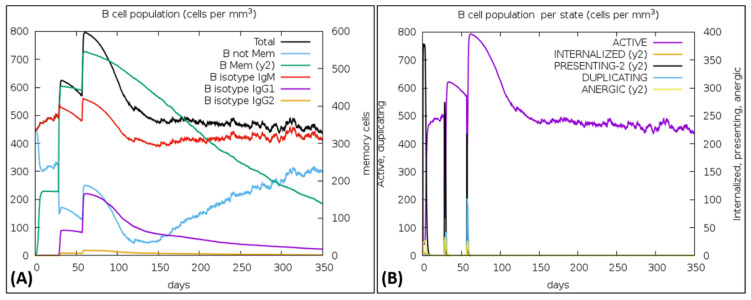
(**A**) Showed the increase in B-cells, (**B**) Rapid peak rise shows an increase in active B-cells.

**Figure 9 vaccines-10-00869-f009:**
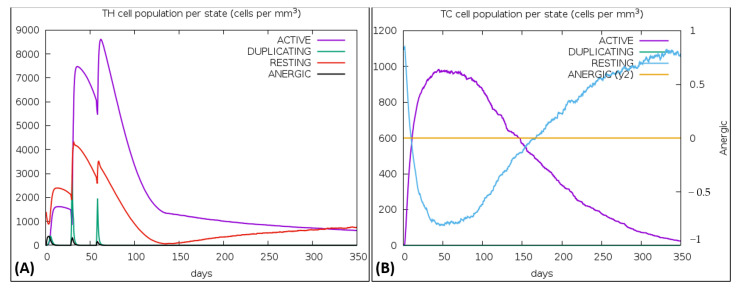
(**A**) Representation of regulatory T-cell production in immune response, (**B**) Graph showing the sharp rise in Cytotoxic T-cells.

**Figure 10 vaccines-10-00869-f010:**
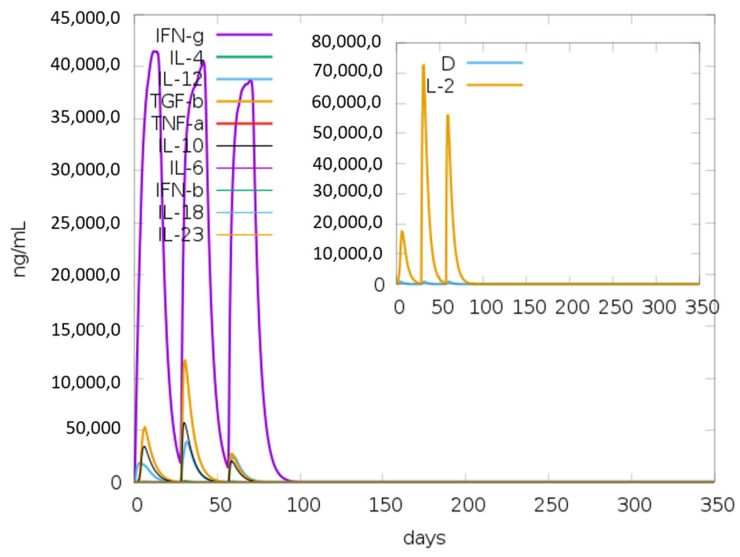
Graphical representation about the production of various immunogenic molecules such as interferons (IFNs) and interleukins (IL); the small graph has a dark blue line, which shows the dangerous level, but that line is almost straight, showing there is no overreaction of vaccine.

**Table 1 vaccines-10-00869-t001:** Physiochemical properties of HN protein and vaccine predicted by ProtParam.

Molecule	MW (Da)	Instability Index	Half-Life	Theoretical pI	AA No.	GRAVY	Aliphatic Index
HN protein	63,981.34	36.48	30 h	8.17	575	−0.161	94.94
Vaccine	69,661.70	33.32	3.5 h	9.04	641	−0.060	93.53

**Table 2 vaccines-10-00869-t002:** Selected B-cell epitopes for the vaccine.

Start	End	Peptide	Length	Antigenicity Score
343	375	TTPLQGDTKCVTNRCANVNQSVCNDALKITWLK	33	0.5002
445	470	MTIKWAPHEVLSRPGNQDCNWYNRCP	26	0.8932
520	528	RLKNVQLEA	9	1.3398

Allergenicity: Non-Allergen. Toxicity: Non-Toxin.

**Table 3 vaccines-10-00869-t003:** Selected T-cell epitopes for the vaccine.

Start	End	Allele	Peptide	Length	Antigenicity Score
**MHC-I EPITOPES**					
399	407	HLA-A * 26:01	ETIPITQNY	9	0.9982
150	159	HLA-B * 57:01	ISPLDPHDFW	10	0.9645
150	159	HLA-B * 58:01	ISPLDPHDFW	10	0.9538
94	102	HLA-A * 68:02	EVISRTINI	9	0.9457
247	256	HLA-A * 68:01	HTYDINDNRK	10	0.9373
305	313	HLA-A * 24:02	RYKNEDITF	9	0.9317
41	49	HLA-B * 57:01	TTMHTILSF	9	0.9105
389	397	HLA-A * 02:03	YLSDRPKIV	9	0.8992
305	313	HLA-A * 23:01	RYKNEDITF	9	0.8926
364	373	HLA-B * 57:01	VCNDALKITW	10	0.8925
2	11	HLA-B * 44:03	AEKGKTNSSY	10	0.8767
2	11	HLA-B * 44:02	AEKGKTNSSY	10	0.8757
150	159	HLA-B * 53:01	ISPLDPHDFW	10	0.8588
367	375	HLA-A * 68:01	DALKITWLK	9	0.8520
41	49	HLA-B * 58:01	TTMHTILSF	9	0.8503
290	298	HLA-A * 68:01	LVFDILDLK	9	0.8461
**MHC-II EPITOPES**					
46	60	HLA-DRB4 * 01:01	ILSFIIMILCIDLII	14	0.08
45	59	HLA-DRB4 * 01:01	TILSFIIMILCIDL	14	0.08
96	110	HLA-DRB4 * 01:01	ISRTINIQSSVQSGI	14	0.15
95	109	HLA-DRB4 * 01:01	VISRTINIQSSVQSG	14	0.2
94	108	HLA-DRB4 * 01:01	EVISRTINIQSSVQS	14	0.23
166	180	HLA-DRB3 * 02:02	PLLSNNPNISLLPGP	14	0.46
44	58	HLA-DRB4 * 01:01	HTILSFIIMILCIDL	14	0.76
514	528	HLA-DRB4 * 01:01	EIINMLRLKNVQLEA	14	1.2
515	529	HLA-DRB4 * 01:01	IINMLRLKNVQLEAA	14	1.2
516	530	HLA-DRB4 * 01:01	INMLRLKNVQLEAAY	14	1.4
415	429	HLA-DRB1 * 15:01	LKLGKKIYIYTRSSG	14	1.5
43	57	HLA-DRB1 * 15:01	MHTILSFIIMILCID	14	1.6
43	57	HLA-DRB4 * 01:01	MHTILSFIIMILCID	14	2.3
517	531	HLA-DRB4 * 01:01	NMLRLKNVQLEAAYT	14	2.6
511	525	HLA-DRB4 * 01:01	NTSEIINMLRLKNVQ	14	2.9

Allergenicity: Non-Allergen. Toxicity: Non-Toxin. * Proposed peptides.

## Data Availability

More data related to this study can be accessed by sending a reasonable email to dr.naveed@ucp.edu.pk.

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
