# Peer review of "Execution and Design of an Anti HPIV-1 Vaccine with Multiple Epitopes Triggering Innate and Adaptive Immune Responses: An Immunoinformatic Approach"

_vaccines, 2022, doi:10.3390/vaccines10060869_

Round 1

Reviewer 1 Report

The authors describe a bioinformatic work to design a multi-epitope assembly for Hemagglutinine-neuraminidase protein of HPIV-a.

The work in interesting and innovative, but lacks some methodological details. I suggest to includes required details before the publication.

Minor revisions:

  • Title 2.12: I suggest modifying the paragraph title in “Molecular docking and dynamics”. Moreover, include more details here or in the corresponding results section (3.12, 3.13), such as molecules employed for docking and MD, MD duration, solvation details, etc.
  • Figure 1: please organize the figure in the correct way: A, B, C and D.
  • Paragraph 3.2: what score is reported? What tool is employed?
  • Table 2 and table 3: column “Allergenicity” and “Toxicity” can probably be removed, and “non-allergen” and “non-toxin” can be added to table caption.
  • Line 224: molecule formula is not informative, please remove it.
  • Paragraph 3.9: Have the authors verified if the single epitope conformation (3D structure) is maintained in the multi-epitope construct?
  • Figure 5: Please shortly discuss figure 5.
  • Paragraph 3.14: I don’t understand why the authors design a vector to experimentally produce the vaccine if they don’t use it.
  • Line 334: Please substitute Li, Fan, Lai, etc. with Li et al.

Author Response

Reviewer 1

Comments and Suggestions for Authors

The authors describe a bioinformatic work to design a multi-epitope assembly for Hemagglutinine-neuraminidase protein of HPIV-a. The work in interesting and innovative, but lacks some methodological details. I suggest to includes required details before the publication.

Minor revisions:

  • Title 2.12: I suggest modifying the paragraph title in “Molecular docking and dynamics”. Moreover, include more details here or in the corresponding results section (3.12, 3.13), such as molecules employed for docking and MD, MD duration, solvation details, etc.

Response: (Line 162, 289-300, 306-315) The title for section 2.12 has been modified accordingly. Furthermore, the section 3.12 and 3.13 has been elaborated further.

  • Figure 1: please organize the figure in the correct way: A, B, C and D.

Response: Figure 1 has been corrected.

  • Paragraph 3.2: what score is reported? What tool is employed?

Response: (Line199-209) Section 3.2 has been elaborated accordingly. The details about the tool have been provided. Furthermore, in case of B-cell-restricted epitope Predictions, instead of score value, there was threshold value, which has been mentioned in the revised version of manuscript.

  • Table 2 and table 3: column “Allergenicity” and “Toxicity” can probably be removed, and “non-allergen” and “non-toxin” can be added to table caption.

Response: The respective columns has been removed and the information has been added in the table caption accordingly.

  • Line 224: molecule formula is not informative, please remove it.

Response: Removed.

  • Paragraph 3.9: Have the authors verified if the single epitope conformation (3D structure) is maintained in the multi-epitope construct?

Response: Dear Reviewer, we would like to appreciate your query regarding the confirmation to maintain the 3D structure of single epitope conformation into the multiple epitope construct. Unfortunately, we didn’t find any bioinformatics tool to confirm this query. For 3D structure, we already verified it using PROCHECK (Ramachandran Plot) and ProSA web (3.10 Tertiary Structure Refinement and Validation)

  • Figure 5: Please shortly discuss figure 5.

Response: Figure 5 has been elaborated in section 3.13 of the revised version of manuscript.

  • Paragraph 3.14: I don’t understand why the authors design a vector to experimentally produce the vaccine if they don’t use it.

Response: (Line 318) The heading 3.14 has been corrected as “Adaptation and enhancement of codon along with in-silico cloning. Also, the legend for figure 3 has been corrected as “In-silico replication of the pET28a (+) plasmid using SnapGene software to construct a recombinant plasmid.” Furthermore, the paragraph has been revised. The vaccine construct is there in the Plasmid, for the convenience we can label it this time (Figure 7).

  • Line 334: Please substitute Li, Fan, Lai, etc. with Li et al.

Response: Corrected.

Reviewer 2 Report

The authors designed multiple epitope-based vaccines against HPIV-1 by taking viral HN protein as a probable vaccine candidate. And the authors selected multiple epitopes in accordance with their allergenicity, antigenicity and toxicity scoring. The reviewer has a few comments which the authors may address: 

  1. What is the different effect of MHC-I and for MHC-II 15 epitopes on predicting T-cells restricted epitopes and B-cell-restricted epitopes for HN proteins? The authors should give their interpretations and discuss the implications.
  2. What is the function of the adjuvant sequence on a designed vaccine? The authors need to introduce this information in the part of “Introduction”. Whether adjuvant sequence will influence the quality of the designed vaccine? The authors need to discuss it in depth.
  3. The authors talked about vaccine stability and showed the original structure of the designed vaccine. What influences vaccine stability? How about the stability of the designed vaccine and how will the authors confirm it when this vaccine is applied to the patients?
  4. Were there any vaccines for HPIV-1? The authors need to introduce the information. And what are the merits and demerits of the designed vaccines compared with other vaccines? The authors should acknowledge the limitations and state their recommendations.
  5. Please correct the word 'Immunoinformatics' as 'Immunoinformatic' in the Title.

Author Response

Reviewer 2

Comments and Suggestions for Authors

The authors designed multiple epitope-based vaccines against HPIV-1 by taking viral HN protein as a probable vaccine candidate. And the authors selected multiple epitopes in accordance with their allergenicity, antigenicity and toxicity scoring. The reviewer has a few comments which the authors may address: 

  1. What is the different effect of MHC-I and for MHC-II 15 epitopes on predicting T-cells restricted epitopes and B-cell-restricted epitopes for HN proteins? The authors should give their interpretations and discuss the implications.

Response: (Line 199-223, 377-382) The different effect of MHC-I and MHC-II 15 epitopes on predicting T-cells restricted epitopes and B-cell-restricted epitopes for HN proteins has been given in section 3.2 and 3.3 respectively. Furthermore, the following paragraph has been added in the discussion section of revised manuscript.

“MHC-I and MHC-II are involved in adaptive immune response where these peptides present the antigen to CD4+ (cytotoxic T-cells) and CD8+ (helper T-cells) respectively (Wieczorek et al., 2017). While the B-cell epitopes are recognized by the B-lymphocytes (via B-cell receptors) (Adler et al., 2017). So, in the current study, we utilized the combination of MHC-I and MHC-II T-cell restricted epitopes and B-cell cell restricted epitopes of HN-protein for the development of strong immunity.”

  1. What is the function of the adjuvant sequence on a designed vaccine? The authors need to introduce this information in the part of “Introduction”. Whether adjuvant sequence will influence the quality of the designed vaccine? The authors need to discuss it in depth.

Response: (In Introduction line 83-90): A paragraph has been added in the Introduction section of revised manuscript.

“The establishment of epitope-based vaccines involves the attachment of predicted epitopes and adjuvant through linkers. In epitope-based vaccines, the antigenic effect is limited as compared to live attenuated vaccines. In case of purified epitope vaccine, the adjuvant is an essential component that can enhance the magnitude of antigenicity of antigens (epitopes) in adaptive immune responses. This magnification of antigens by Adjuvant can enhance antibody production and affects the longevity of the vaccine through co-administration of epitopes. A limited number of reported Adjuvants are available that are incorporated into human vaccines [18].”

In the discussion section, Line 383-393, describes the different adjuvants in epitope vaccine designs.

  1. The authors talked about vaccine stability and showed the original structure of the designed vaccine. What influences vaccine stability? How about the stability of the designed vaccine and how will the authors confirm it when this vaccine is applied to the patients?

Response: The designed vaccine construct (protein) is stable as per the Physiochemical properties.  In table 1 the score of the Instability Index is 33.32 as per protparam the construct is stable. Secondly in 2.10 and 3.11 implication of the Disulphide tool on Design 2.0 (DbD2) to identify the disulphide linkages is the other factor that controls the stability of the construct’s 3D conformation.

Unfortunately, at this stage we cannot confirm the stability od vaccine on the patients without the supporting data from In-vitro and in-vivo trials. In the current study, we can just provide the data to prove the fact that the vaccine construct is stable.

  1. Were there any vaccines for HPIV-1? The authors need to introduce the information. And what are the merits and demerits of the designed vaccines compared with other vaccines? The authors should acknowledge the limitations and state their recommendations.

Response: (Line 405-412) The study limitations and recommendations has been written separately in the revised version of manuscript. Furthermore, some points about the vaccine has been elaborated in the revised version of manuscript.

  1. Please correct the word 'Immunoinformatics' as 'Immunoinformatic' in the Title.

Response: Corrected.

Reviewer 3 Report

  1. Abstact and conclusion should be rewritten.
  2. Discussion should be impoved.
  3. Physicochemical properties are not presented fully.
  4. Molecular docking is poor.

Author Response

Reviewer 3

Comments and Suggestions for Authors

  1. Abstact and conclusion should be rewritten.

Response: (Line 36, 38, 45, 46-53) The Abstract and conclusion sections has been revised accordingly.

  1. Discussion should be impoved.

Response: (Line 346-353, 359-363, 368-370, 377-395) The discussion section has been thoroughly revised.

  1. Physicochemical properties are not presented fully.

Response: (Line 258-268) The section 3.8 for physiochemical properties has been elaborated in the revised version of manuscript.

  1. Molecular docking is poor.

Response: (Line 289-304) The molecular docking section has been revised and rerun again. The figure 4 for docking results has also been updated in the revised version of manuscript.

Reviewer 4 Report

I have reviewed the manuscript “Execution and design of an anti HPIV-1 vaccine with multiple epitopes triggering innate and adaptive immune responses: An Immunoinformatics approach” submitted to “Vaccines Journal” for publication. In this paper, authors have designed a multiple epitope-based vaccine against HPIV-1 by taking viral HN protein as a probable vaccine candidate. In addition, the multiple epitopes were selected in accordance of their allergenicity, antigenicity and toxicity scoring. I found this work interesting and fit well within the scope of this journal. The manuscript needs some major improvements; there are a few suggestions that authors may consider to improve it further:

The use of the English language is reasonable, however, there are a number of punctuation and grammatical errors; that should be corrected and rephrased using academic.

Abstract: is precisely written, and the aim of the study is mentioned. Please include some more information about the results/finding to enhance the impact of this section.

The introduction; is detailed, compact, covering the background information and the rationale of the study effectively.

References citations are not in order, the introduction starts with ref no 14.

Authors should define all the abbreviations at their first appearance in the text as many abbreviations used are not defined.

All the data and Figures are presented logically, and findings are described in the text.

Furthermore, In discussion section, please correct the format of using “et al”; as last name followed by et al. for example line 343, ref 47 should be corrected. Similarly, line 323 should be corrected as Qamar, et al.

The limitations of the study should be further elaborated.

Author Response

Reviewer 4

Comments and Suggestions for Authors

I have reviewed the manuscript “Execution and design of an anti HPIV-1 vaccine with multiple epitopes triggering innate and adaptive immune responses: An Immunoinformatics approach” submitted to “Vaccines Journal” for publication. In this paper, authors have designed a multiple epitope-based vaccine against HPIV-1 by taking viral HN protein as a probable vaccine candidate. In addition, the multiple epitopes were selected in accordance of their allergenicity, antigenicity and toxicity scoring. I found this work interesting and fit well within the scope of this journal. The manuscript needs some major improvements; there are a few suggestions that authors may consider to improve it further:

The use of the English language is reasonable, however, there are a number of punctuation and grammatical errors; that should be corrected and rephrased using academic.

Response: The manuscript has been revised thoroughly for English proofreading and grammatical mistakes.

Abstract: is precisely written, and the aim of the study is mentioned. Please include some more information about the results/finding to enhance the impact of this section.

Response: (Line 36, 38, 45, 46-53) The Abstract has been revised accordingly.

The introduction; is detailed, compact, covering the background information and the rationale of the study effectively.

Response: Dear reviewer we would like to appreciate your keen review of current manuscript.

References citations are not in order, the introduction starts with ref no 14.

Response: The reference citations has been corrected in accordance with their order.

Authors should define all the abbreviations at their first appearance in the text as many abbreviations used are not defined.

Response: All of the abbreviations has been carefully checked and described at their first appearance.

All the data and Figures are presented logically, and findings are described in the text.

Response: Dear reviewer we would like to appreciate your keen review of current manuscript.

Furthermore, In discussion section, please correct the format of using “et al”; as last name followed by et al. for example line 343, ref 47 should be corrected. Similarly, line 323 should be corrected as Qamar, et al.

Response: Corrected.

The limitations of the study should be further elaborated.

Response: (Line 405-412) The study limitations has been written separately in the revised version of manuscript.

Round 2

Reviewer 2 Report

In the revised article, the authors modified the manuscript referred to the comments, and answered the questions comprehensively.

Reviewer 3 Report

Authors revised manuscript and paper is suitable for publication.

Reviewer 4 Report

The authors' responses and revision of the manuscript are satisfactory.